# Root Characteristics for Maize with the Highest Grain Yield Potential of 22.5 Mg ha$^{-1}$ in China

**Long Zhang** [1,2,†], **Guangzhou Liu** [3,4,†], **Yunshan Yang** [5], **Xiaoxia Guo** [5], **Shuai Jin** [1], **Ruizhi Xie** [2], **Bo Ming** [2], **Jun Xue** [2], **Keru Wang** [2], **Shaokun Li** [2,*] and **Peng Hou** [1,*]

1   College of Agronomy, Ningxia University, Yinchuan 750021, China
2   Key Laboratory of Crop Physiology and Ecology, Institute of Crop Sciences, Beijing 100081, China
3   State Key Laboratory of North China Crop Improvement and Regulation, College of Agronomy, Hebei Agricultural University, Baoding 071001, China
4   Key Laboratory of Crop Growth Regulation of Hebei Province, College of Agronomy, Hebei Agricultural University, Baoding 071001, China
5   The Key Laboratory of Oasis Eco-Agriculture, Xinjiang Production and Construction Corps, College of Agronomy, Shihezi University, Shihezi 832000, China
*   Correspondence: lishaokun@caas.cn (S.L.); houpeng@caas.cn (P.H.); Tel./Fax: +86-10-82108891 (S.L.); Tel./Fax: +86-10-82108595 (P.H.)
†   These authors contributed equally to this work.

**Abstract:** In maize (*Zea mays* L.), rational root structure promotes high grain yield under dense sowing conditions. This study was conducted at Qitai Farm in Xinjiang, China, in 2019 and 2021. A traditional wide and narrow row planting method was adopted, with wide rows of 0.7 m and narrow rows of 0.4 m. The cultivars DH618 and SC704, which have grain yield potentials of 22.5 and 15 Mg ha$^{-1}$, respectively, were selected for study of the root structure and distribution characteristics under high-yield and high-density planting conditions. The highest yield (20.24 Mg ha$^{-1}$) was achieved by DH618 under a planting density of $12 \times 10^4$ plants ha$^{-1}$. The root structure of DH618 was well developed at that planting density, and the root dry weight (RDW) was 17.49 g plant$^{-1}$ and 14.65 g plant$^{-1}$ at the silking and maturity stages, respectively; these values were 7.56% and 11.86% higher, respectively, than those of SC704. At the silking stage, the proportions of RDW at soil depths of 0–10, 10–20, 20–40, and 40–60 cm were 66.29%, 11.83%, 16.51%, and 5.38%, respectively, for DH618; over the 20–60 cm soil layer, this was an average of 4.04% higher than the RDW of SC704. At maturity, the proportions of RDW at soil depths of 0–10, 10–20, 20–40, and 40–60 cm were 61.40%, 11.19%, 17.19%, and 10.21%, respectively, for DH618, which was an average of 9.59% higher than that of SC704 over the 20–60 cm soil layer. At maturity, DH618 roots were mainly distributed in the narrow rows, accounting for 72.03% of the root structure; this was 9.53% higher than the roots of SC704. At silking and maturity, the root weight densities of DH618 were 471.98 g m$^{-3}$ and 382.98 g m$^{-3}$, respectively (5.18% and 5.97% higher, respectively, than the root weight densities of SC704). The root lengths of DH618 were 239.72 m plant$^{-1}$ and 199.04 m plant$^{-1}$ at the silking and maturity stages, respectively; these were 16.45% and 25.39% higher, respectively, than the root lengths of SC704. The root length densities were 0.58 cm cm$^{-3}$ and 0.46 cm cm$^{-3}$ at the silking and maturity stages, respectively, and these were 16.86% and 17.08% higher, respectively, than the root length densities of SC704. This study indicated that the maize hybrid DH618 had a more developed root structure with increased root distribution in the deep soil and narrow rows under high-density planting compared to cultivar SC704, contributing to high grain yield under dense planting.

**Keywords:** maize; high grain yield potential; root dry weight; root length; root length density

## 1. Introduction

Maize is the most widely planted food crop in the world [1] due to its high adaptability to different growth environments. It is not only an important food crop but also an impor-

tant feed crop. At present, the world is facing unprecedented challenges; the per-capita cultivated land area is decreasing every year, meaning that it is increasingly important to ensure food security [2]. Improvement of per-unit maize yield within the limited cultivated land area available is an important component in alleviating the global food crisis [3].

Increasing planting density is a key measure by which maize yield can be improved [4,5]. It is also a critical technical measure that is taken to establish high maize yield records in China [5]. However, when planting density is increased, the light-receiving conditions in the canopy are changed [6]. Many studies have been carried out to determine methods of effectively improving light interception and usage in plants. These studies have addressed parameters such as plant type [7,8] and source-sink relationships [9], and improvements have been made in those areas as a result. Research has shown that the high grain yield potential of modern maize hybrids is not only related to canopy structure, but also to root distribution in the soil [10,11].

As the connector between plants and the soil, roots are the key site for anchoring plants and absorbing nutrients [11,12]. Studies have shown that greater root dry weight (RDW) in maize can provide increased soil resources to the plant to maintain growth and development [13,14]. Root structure plays an important role in obtaining soil resources to promote plant growth and yield formation [10,11]. The soil resources needed by plants are typically distributed in a highly heterogeneous pattern, which promotes a large degree of "developmental plasticity" in the root structure [15]. The distribution of root length (RL) and RL density (RLD) in space are key indexes in studying root structure [16,17]. For example, distribution of a large number of roots in the deep-soil layer can allow increased nitrogen capture [12,18], whereas roots distributed between plant rows can effectively avoid inter-plant competition for soil resources [14]. Many studies have shown that improvements in root structure can promote increased crop yield [11,15,17,19].

The high grain yield potential of many modern maize hybrids is reportedly related to root distribution. Under limited light conditions, building efficient roots is the key to obtaining soil resources [10]. From a population perspective, the spatial distribution of roots under high-density maize planting promotes effective soil exploration, and the limited photosynthetic products and lower metabolic consumption in roots jointly increase the potential for high grain yield [11,19]. Absorption of specific nutrients by the roots strongly depends on the plant genotype [20]. The root structure and distribution characteristics significantly differ between accessions grown under the same environmental conditions, and the synergistic or antagonistic effects of various hormones on the growth of different root types are also complex and diverse [15]. Modulation of root structure and distribution characteristics is an important component of optimizing grain yield potential, and research into root characteristics is thus the key to improving maize yield. Hammer et al. [21] showed that changes in maize root structure and water capture have a direct impact on biomass accumulation and historical yield trends in the United States. However, no research has been conducted in varieties with a grain yield potential of 22.5 Mg ha$^{-1}$.

The objective of this study was to explore the root characteristics of high-yield maize, specifically maize with a grain yield potential of 22.5 Mg ha$^{-1}$. The results will supplement the known characteristics of maize cultivars with a grain yield potential of 22.5 Mg ha$^{-1}$ and contribute to the breeding and cultivation of high-yield maize cultivars in the future.

## 2. Materials and Methods

### 2.1. Experimental Design

This study was carried out at Qitai Farm in Xinjiang, China (89°34′ E, 43°12′ N) in the same field in 2019 and 2021. The traditional wide and narrow row planting method was used, with wide rows of 0.7 m and narrow rows of 0.4 m. To explore the spatial distribution and morphological characteristics of high-yield maize root structure, the cultivars Denghai 618 (DH618) and SC704, with grain yield potentials of 22.5 Mg ha$^{-1}$ and 15 Mg ha$^{-1}$, respectively, which were obtained at their optimum densities during

a long-term high yield exploration, were selected. Both were planted at a density of $7.5 \times 10^4$ (D1) and $12.0 \times 10^4$ plants ha$^{-1}$ (D2).

The two sowing dates were 20 April 2019, and 17 April 2021. The two sampling dates were 17 July, 1 October 2019, and 16 July, 26 September 2021, and the two harvest dates were 3 October 2019, and 28 September 2021. Growing degree days (GDDs) of DH618 and SC704 were 1501.7 °C and 1533.6 °C in the two experiment years. Drip water irrigation (15 mm) was applied in the 24 h after sowing to ensure uniform seedling emergence. After emergence, seedlings were not irrigated until the jointing stage to promote root growth. After the jointing stage, drip irrigation technology was used to conduct integrated quantitative water every 9–10 days. The total amount of water applied was approximately 540 mm per growing season. The physical and chemical properties of the soil at the test site were assessed at a depth of 0–60 cm (Table 1). To ensure soil basic fertility, consistent with the target yield, 150 kg ha$^{-1}$ N fertilizer (urea), 225 kg ha$^{-1}$ P fertilizer (super phosphate), and 75 kg ha$^{-1}$ K fertilizer (potassium sulfate) were applied as base fertilizer, and a total of 300 kg ha$^{-1}$ N fertilizer (urea) was applied as topdressing fertilizer during the growing season. Meteorological data were captured at meteorological observation points near the test site during the 2019 and 2021 maize growing seasons (Table 2). During this period, good management measures were maintained in the field. Sufficient water and nutrients were provided throughout the growth period to avoid water or nutrient deprivation stress, and diseases, insects, and weeds were well-controlled.

**Table 1.** Physical and chemical properties of test site soil at a depth of 0–60 cm.

| Year | Organic Matter (g kg$^{-1}$) | Alkaline N (mg kg$^{-1}$) | Olsen P (mg kg$^{-1}$) | Available K (mg kg$^{-1}$) | Bulk Density (g cm$^{-3}$) | pH |
|---|---|---|---|---|---|---|
| 2019 | 14.1 | 87.6 | 53.8 | 108.6 | 1.37 | 7.97 |
| 2021 | 17.8 | 78.0 | 66.3 | 330.1 | 1.34 | 7.75 |

**Table 2.** Maximum temperature ($T_{max}$), minimum temperature ($T_{min}$), diurnal temperature variation ($T_d$), daily solar radiation (Sr), and accumulated precipitation (Pre) during the maize growing seasons in 2019 and 2021.

| Year | Tmax (°C) | Tmin (°C) | Td (°C) | Sr (M Jm$^{-2}$ day$^{-1}$) | Pre (mm) |
|---|---|---|---|---|---|
| 2019 | 27.1 | 12.3 | 14.8 | 9.8 | 189.1 |
| 2021 | 28.5 | 10.1 | 18.4 | 9.4 | 159.8 |

*2.2. Sampling and Measurements*

2.2.1. Root Sampling Method

Three successive and representative plants of uniform size with similar growth rates were selected, and the aboveground tissues were removed. The soil for each plant was then divided into three horizontal portions (8 and 5 cm in width for samples grown under D1 and D2 conditions, respectively) based on the standard of maize plant spacing, and the soils were divided into five portions (11 cm in length) based on the average row spacing (Figure 1). Soil samples were divided vertically into four layers of 0–10, 10–20, 20–40, and 40–60 cm, with 15 soil samples for each plant in each layer. The soil volume in each sample was $8/5 \times 11 \times 10$ cm$^3$ for the 0–10 and 10–20 cm soil layers and $8/5 \times 11 \times 20$ cm$^3$ for the 20–40 and 40–0 cm soil layers. In the process of soil layering, the connected roots were removed with scissors. The roots and soil were placed into mesh bags together and washed with water to remove dead roots and other impurities. The roots were then placed in the refrigerator for the next operation.

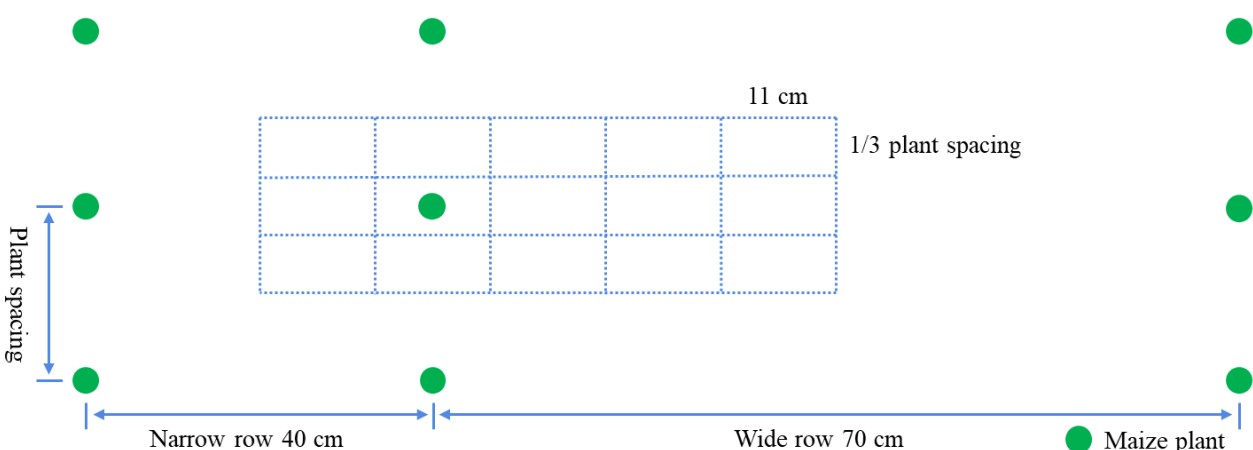

**Figure 1.** Schematic diagram of single maize plant root sampling.

### 2.2.2. Root Index Analysis Methods

A scanner (Epson V800, Indonesia) was used to scan each processed root, and the resulting images were analyzed with WinRhizo Pro Vision5.0 (Canada) to determine the total root length (RL). Samples were dried at 105 °C for 30 min and then at 80 °C to a constant weight, which was calculated as the total root dry weight (RDW). Root weight density (RWD) was calculated as RDW divided by soil volume (SV), and root total length density (RLD) was calculated as RL divided by SV.

### 2.2.3. Grain Yield Determination Method

At physiological maturity, ears from a central 5 m $\times$ 2 row area were harvested in each plot, and ten ears were selected according to the average ear weight and manually threshed from each area. After recording the kernel weight, the moisture content was measured using a PM-8188 portable grain moisture meter (Kett), and the final grain yield was standardized at 14.0% moisture content.

### 2.3. Statistical Analysis

Data were statistically analyzed and plotted in Origin (2022). The data sources of this paper were analyzed by one-way and three-way ANOVA followed by the LSD test to compare the mean values among the treatments at $p < 0.05$.

## 3. Results

### 3.1. Grain Yield under Different Planting Densities

On average over the two growth seasons, DH618 had a higher grain yield under D2 than D1 conditions; this was significantly higher than the yield of SC704 at either planting density. The average grain yields of DH618 were 19.57 Mg ha$^{-1}$ and 20.24 Mg ha$^{-1}$ under D1 and D2 density, and the SC704 were 17.19 Mg ha$^{-1}$ and 14.94 Mg ha$^{-1}$, respectively. For D1 and D2 conditions, there were significant yield differences for DH618 in 2019 and for SC704 in 2021, but no significant differences for SC704 in 2019 or for DH618 in 2021 (Figure 2).

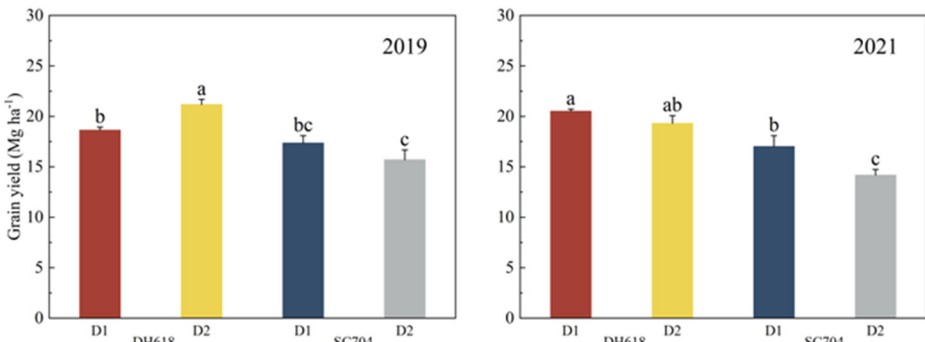

**Figure 2.** Grain yield for DH618 and SC704 under two planting densities, D1 ($7.5 \times 10^4$) and D2 ($12.0 \times 10^4$). Means in the same year not sharing any letter above the bars are significantly different by the LSD-test at the 5% level of significance.

*3.2. Root Dry Weight*

At the silking stage in 2019, under D1 and D2 conditions, the average RDW was 18.87 g plant$^{-1}$ and 16.08 g plant$^{-1}$, respectively, for DH618, and 17.61 g plant$^{-1}$ and 15.65 g plant$^{-1}$, respectively, for SC704; at maturity, the average RDW under D1 and D2 conditions was 17.47 g plant$^{-1}$ and 14.48 g plant$^{-1}$, respectively, for DH618, and 17.55 g plant$^{-1}$ and 12.74 g plant$^{-1}$, respectively, for SC704. In 2021, under D1 and D2 conditions, the average RDW was 21.81 g plant$^{-1}$ and 18.90 g plant$^{-1}$, respectively, for DH618, and 22.50 g plant$^{-1}$ and 16.87 g plant$^{-1}$, respectively, for SC704; at maturity, the average RDW was 16.89 g plant$^{-1}$ and 14.81 g plant$^{-1}$, respectively, for DH618, and 18.85 g plant$^{-1}$ and 13.44 g plant$^{-1}$, respectively, for SC704.

In 2019 and 2021, the roots of cultivars DH618 and SC704 were mainly distributed in the 0–20 cm soil layer at both the silking and maturity stages, accounting for ~68.92–83.84% of all roots (Figure 3). Under D1 conditions, the distribution of RDW for DH618 at the silking stage in the 0–10, 10–20, 20–40, and 40–60 cm soil layers was 64.70%, 15.72%, 13.33%, and 6.25%, respectively; under D2 conditions, the distribution was 66.29%, 11.83%, 16.51%, and 5.38%, respectively. Under D1 conditions, the distribution of RDW for SC704 in the 0–10, 10–20, 20–40, and 40–60 cm soil layers was 64.69%, 12.46%, 13.83%, and 9.03%, respectively, and under D2 conditions the distribution was 70.54%, 11.63%, 11.71%, and 6.13%, respectively. At maturity, the distribution of RDW for DH618 in the 0–10, 10–20, 20–40, and 40–60 cm soil layers was 69.19%, 10.05%, 14.37%, and 6.39%, respectively, under D1 conditions, and 61.40%, 11.19%, 17.19%, and 10.21%, respectively, under D2 conditions; for SC704, the RDW distribution in the 0–10, 10–20, 20–40, and 40–60 cm soil layers was 67.47%, 12.39%, 13.89%, and 6.25%, respectively, under D1 conditions, and 72.52%, 9.67%, 11.19%, and 6.63%, respectively, under D2 conditions.

Compared with the silking stage, the average single-plant RDW was decreased at maturity. Under D2 conditions, RDW was an average of 19.42% and 24.19% lower in DH618 and SC704, respectively, at maturity compared to the silking stage. On average over the two experimental years, RDW was 16.80% and 31.20% lower in DH618 and SC704, respectively, under D2 compared to D1 conditions. This indicated that RDW was reduced with increased planting density, and that RDW also decreased as the growth period advanced.

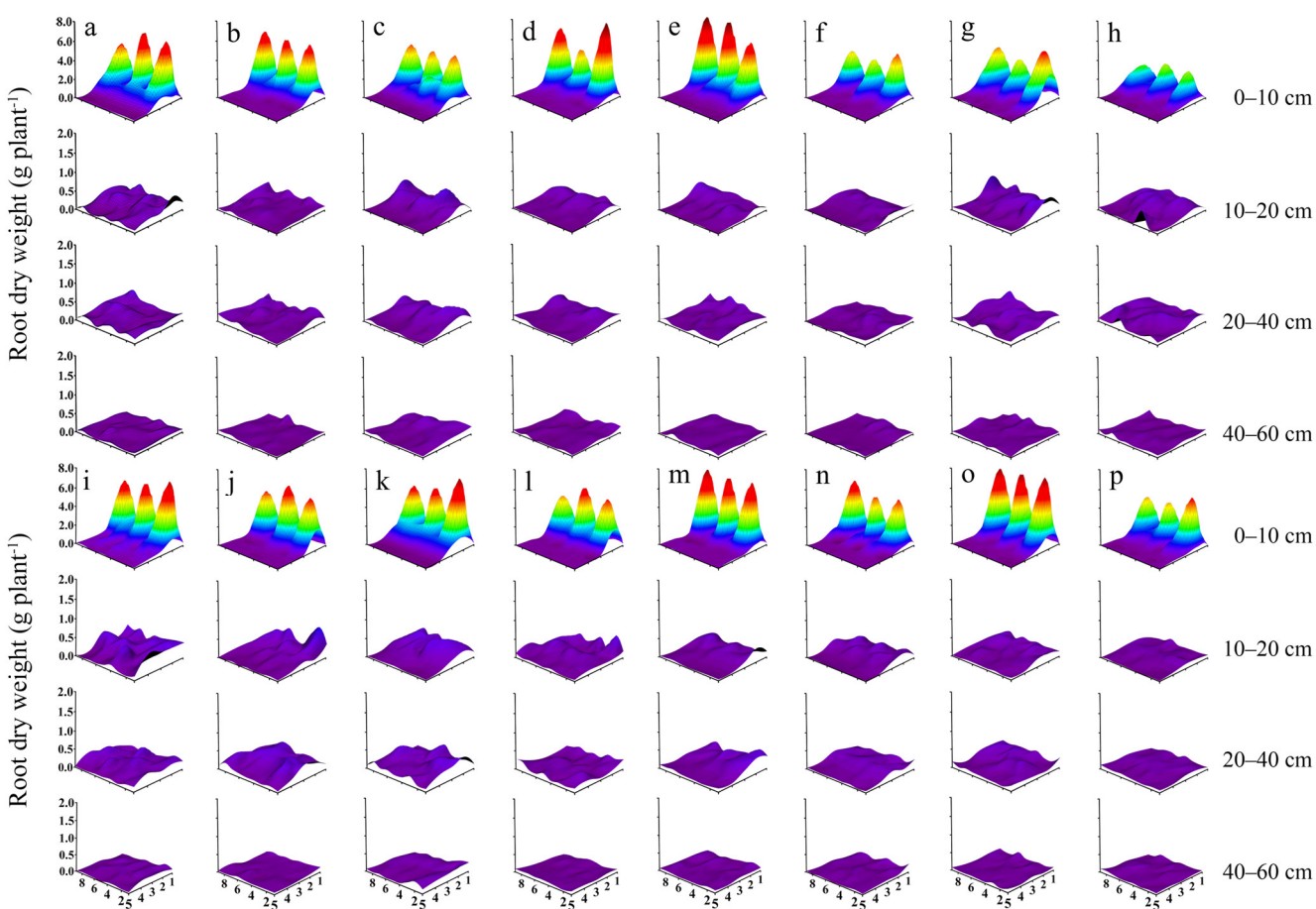

Distance (8 cm/ 5 cm)/ Distance (11 cm)

**Figure 3.** Vertical distribution of root dry weight (RDW) of two maize cultivars (DH618 and SC704) grown at two planting densities (D1 and D2). (**a–d**) RDW at the silking stage in 2019 for (**a**) DH618 grown at D1, (**b**) DH618 grown at D2, (**c**) SC704 grown at D1, and (**d**) SC704 grown at D2. (**e–h**) RDW at maturity in 2019 for (**e**) DH618 grown at D1, (**f**) DH618 grown at D2, (**g**) SC704 grown at D1, and (**h**) SC704 grown at D2. (**i–l**) RDW at the silking stage in 2021 for (**i**) DH618 grown at D1, (**j**) DH618 grown at D2, (**k**) SC704 grown at D1, and (**l**) SC704 grown at D2. (**m–p**) RDW at maturity in 2021 for (**m**) DH618 grown at D1, (**n**) DH618 grown at D2, (**o**) SC704 grown at D1, and (**p**) SC704 grown at D2.

### 3.3. Root Weight Density

In general, RWD decreased along with the increase in soil depth. RWD was also significantly higher under D2 than under D1 conditions for both DH618 and SC704 (Table S1). In 2019, the RWD of DH618 and SC704 at silking was increased by 37.12% and 45.69%, respectively, under D2 compared to D1 conditions; at maturity, RWD increased by 22.84% and 19.41%, respectively. In 2021, the RWD of DH618 and SC704 increased by 34.71% and 23.61%, respectively, at silking, and increased by 40.46% and 13.97%, respectively, at maturity under D2 compared to D1 conditions. There were no significant differences in RWD between DH618 and SC704. Under both density conditions, RWD decreased from silking to maturity, and the difference was significant under D2 conditions.

With respect to soil depth, it was found that the strategies for reducing the RWD of the two cultivars were different from that of the silking stage: on average, under D2 conditions, RWD was decreased in SC704 at maturity compared to the silking stage at the 0–10 cm soil depth, and that at the 10–60 cm soil depth decreased by 37.36%; more roots were retained at the 0–10 cm soil depth. At maturity, RWD of DH618 was decreased by 28.93% in the

0–10 cm soil layer and by 9.21% in the 10–60 cm soil layer compared with the silking stage, indicating that more roots were retained at the 10–60 cm soil depth.

With respect to row distance, the roots of plants in each treatment were primarily concentrated in narrow rows (at 0–22 cm from the center of the narrow rows); RWD was highest within 11–22 cm from the center of the narrow rows. Compared with D1 conditions, RWD was significantly increased in narrow rows under D2 conditions, but there were no significant differences in wide rows (22–55 cm); DH618 showed a significant performance (Table S2). On average, compared with D1 conditions, the RWD of DH618 plants increased by 47.85% in narrow rows and decreased by 0.58% in wide rows under D2 conditions. In contrast, the RWD of SC704 increased by 32.53% in narrow rows and by 8.76% in wide rows under D1 compared to D2 conditions.

As the planting density increased from D1 to D2, there was a corresponding significant increase in RWD (Table S3). In 2019, the average RWD of DH618 and SC704 increased from D1 to D2 by 36.15% and 10.51%, respectively; in 2021, the increases were 46.44% and 33.39%, respectively.

### 3.4. Root Length

In general, RL was significantly higher at the silking stage than at maturity, and RL was significantly higher in DH618 than in SC704 plants (Figure 4). At the silking stage, the total RL values of DH618 and SC704 were 288.70 m and 286.01 m, respectively, under D1 conditions, and 239.72 m and 205.84 m, respectively, under D2 conditions. At maturity, the total RL values of DH618 and SC704 were 271.56 m and 186.34 m, respectively, under D1 conditions, and 199.04 m and 158.75 m, respectively, under D2 conditions. At soil depths of 0–10, 10–20, 20–40, and 40–60 cm, the RL values of DH618 at the silking stage were 135.32 m, 69.31 m, 52.76 m, and 31.30 m, respectively, under D1 conditions, and 88.53 m, 54.97 m, 56.88 m, and 39.34 m, respectively, under D2 conditions. At soil depths of 0–10, 10–20, 20–40, and 40–60 cm, the RL values of SC704 were 124.09 m, 48.97 m, 62.29 m, and 50.66 m, respectively, under D1 conditions, and 87.14 m, 34.93 m, 50.80 m, and 32.97 m, respectively, under D2 conditions. At maturity, the RL values of DH618 at soil depths of 0–10, 10–20, 20–40, and 40–60 cm were 101.22 m, 50.72 m, 70.04 m, and 49.58 m, respectively, under D1 conditions, and 69.19 m, 37.20 m, 67.81 m, and 24.84 m, respectively, under D2 conditions. The RL values of SC704 at soil depths of 0–10, 10–20, 20–40, and 40–60 cm were 73.67 m, 29.21 m, 47.25 m, and 36.22 m, respectively, under D1 conditions, and 79.01 m, 23.11 m, 32.24 m, and 24.38 m, respectively, under D2 conditions.

### 3.5. Root Length Density

In 2019 and 2021, RLD was significantly higher at the silking stage than at maturity, especially under D1 conditions (Table S4). In the silking stage, the average RLD values of DH618 and SC704 over the two years were 0.47 cm cm$^{-3}$ and 0.43 cm cm$^{-3}$, respectively, under D1 conditions, and 0.58 cm cm$^{-3}$ and 0.50 cm cm$^{-3}$, respectively, under D2 conditions. At maturity, under D1 conditions, the average RLD values of DH618 and SC704 over the two years were 0.40 cm cm$^{-3}$ and 0.27 cm cm$^{-3}$, respectively, and under D2 conditions were 0.46 cm cm$^{-3}$ and 0.40 cm cm$^{-3}$, respectively.

With respect to soil depth, the change rule of RLD was consistent across years, growth periods, and density conditions. Overall, RLD was higher at each soil depth under D2 compared to D1 conditions and in DH618 compared to SC704 (Table S4). At soil depths of 0–10, 10–20, 20–40, and 40–60 cm, the RLD values of DH618 were 0.90, 0.45, 0.23, and 0.15 cm cm$^{-3}$, respectively, under D1 conditions, and 0.96, 0.56, 0.38, and 0.19 cm cm$^{-3}$, respectively, under D2 conditions. For SC704, the RLD values at soil depths of 0–10, 10–20, 20–40, and 40–60 cm were 0.75, 0.30, 0.21, and 0.16 cm cm$^{-3}$, respectively, under D1 conditions, and 1.01, 0.35, 0.25, and 0.17 cm cm$^{-3}$, respectively, under D2 conditions.

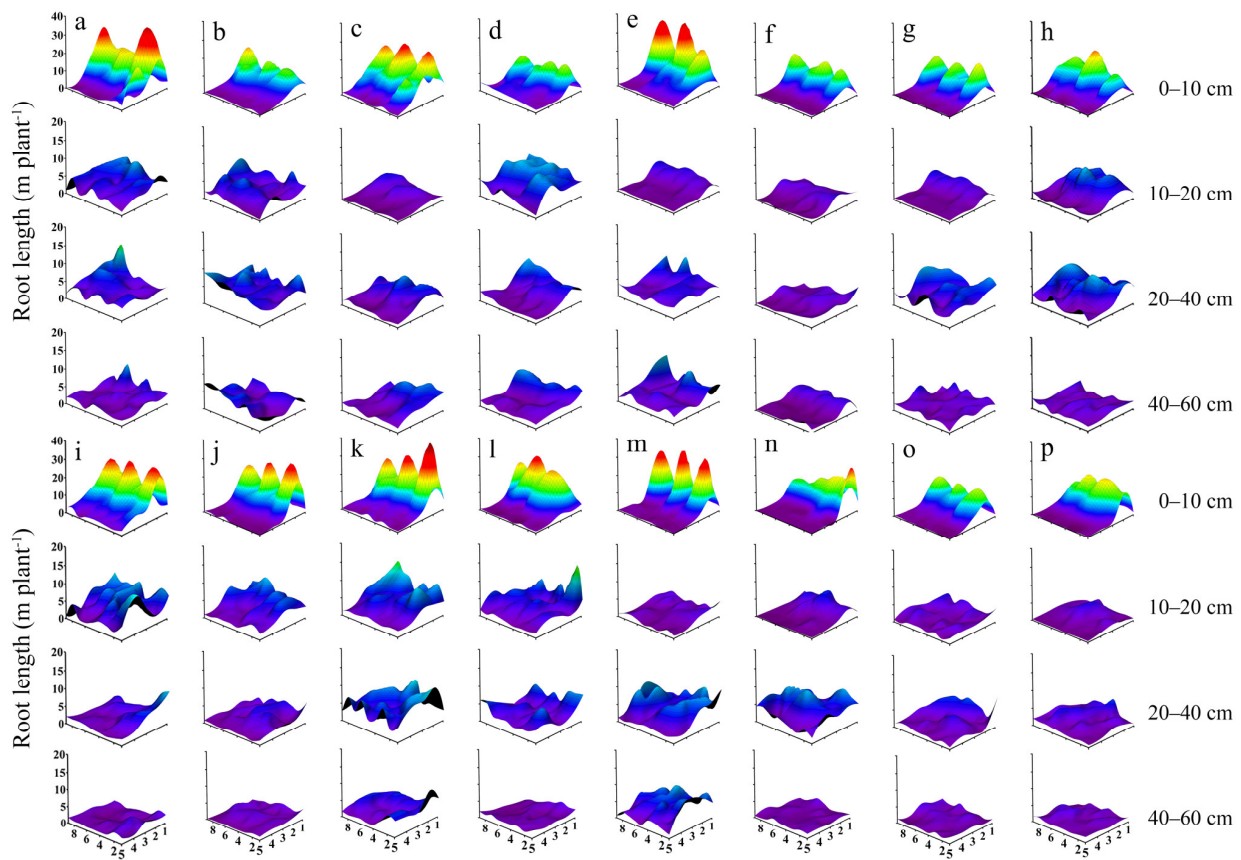

Distance (8 cm/5 cm)/Distance (11 cm)

**Figure 4.** Vertical distribution of root length (RL) for two maize cultivars (DH618 and SC704) grown at two planting densities (D1 and D2). (**a–d**) RL at the silking stage in 2019 for (**a**) DH618 grown at D1, (**b**) DH618 grown at D2, (**c**) SC704 grown at D1, and (**d**) SC704 grown at D2. (**e–h**) RL at maturity in 2019 for (**e**) DH618 grown at D1, (**f**) DH618 grown at D2, (**g**) SC704 grown at D1, and (**h**) SC704 grown at D2. (**i–l**) RL at the silking stage in 2021 for (**i**) DH618 grown at D1, (**j**) DH618 grown at D2, (**k**) SC704 grown at D1, and (**l**) SC704 grown at D2. (**m–p**) RL at maturity in 2021 for (**m**) DH618 grown at D1, (**n**) DH618 grown at D2, (**o**) SC704 grown at D1, and (**p**) SC704 grown at D2.

*3.6. Interaction Effects of Years, Planting Densities, and Cultivars on the Maize Root Characteristics*

There were significant differences in root dry weight and root weight density between the two experiment years as well as the two plant densities (Table 3). Root length and root length density were significantly affected by planting density and cultivar, and interaction of year × cultivar (Table 3).

**Table 3.** ANOVA for the effects of years, planting densities, and cultivars on the maize root characteristics. *, significant at $p < 0.05$; **, significant at $p < 0.01$; ns, no significant difference.

| Sources of Variation | Root Dry Weight | Root Weight Density | Root Length | Root Length Density |
|---|---|---|---|---|
| Year | ** | * | ns | ns |
| Planting density | ** | ** | ** | ** |
| Cultivar | ns | ns | ** | ** |
| Year × Planting density | ns | ns | ns | ns |
| Year × Cultivar | ns | ns | * | * |
| Planting density × Cultivar | ns | ns | ns | ns |
| Year × Planting density × Cultivar | ns | ns | ns | ns |

## 4. Discussion

Root structure is an important contributor to yield, and therefore a focus of grain yield improvement [15,19]. Cultivar DH618 was bred from 521 × DH392 in 2013 [22], which had compact plant type, developed root structure [23], was tolerant to high density [24], and had high grain yield potential [8]. Cultivar SC704 was bred from ZPL773 × ZPL717 and released in China in 1982, which had flat plant type, and was suitable for planting at low density. During our long-term maize high yield exploration, DH618 had the yield potential of 22.5 Mg ha$^{-1}$ and canopy characteristics were already displayed [22,23]. In the present study, DH618 had the average highest grain yield under the higher planting density (D2), which was significantly higher than that of cultivar SC704 (Figure 2) and was similar to the results in our previous studies [22,25].

Previous studies have shown that increases in maize yield are related to higher RDW [14,26]. This is consistent with the results of our study; higher yield in DH618 was associated with larger RDW (Figure 3). We found that maize roots were primarily distributed at a soil depth of 0–20 cm (Figure 3), as in previously published results [26–28]. However, we also found that more roots were distributed in the deep soil (20–60 cm), which could promote increased yield [14,29]. Compared with cultivar SC704, the high-yield variety DH618 had a higher proportion of roots distributed in the deep soil (Figure 3). Increasing the proportion of roots in the deep soil can effectively increase soil exploration capacity and reduce nutrient competition between plants [30]; this trait, which results from selection during the process of variety breeding and improvement, improves the efficiency of nutrient acquisition [14].

This study has shown that RDW was reduced with increased planting density, but RWD significantly increased along with planting density in both DH618 and SC704 plants (Table S1). Previous studies have shown that single-root biomass decreases with increased plant density, whereas the root biomass of the population does not change significantly [10]. This may be related to the planting density of the experiment. When planting density is increased by a reasonable amount, the root density shows a corresponding increase to meet nutrient demands to maintain canopy growth and development. Increases in density can effectively increase the dry weight of the root structure and thus enable the growth and development of larger shoots [24,31]. There are significant positive correlations between RWD and grain yield at the silking and maturity stages [17], consistent with the results of the present study. Root senescence is known to occur from the silking stage through maturity [14,17,26], explaining the decreases in RDW and RWD that we observed between the silking stage and maturity (Table S1). Plants can effectively alleviate the process of senescence by reducing the surface root density, increasing the deep-soil root structure, and exploring, intercepting, and absorbing water and nutrients from the deep soil [10,18,28,29]. DH618 was shown to have good green retention at maturity [23], which may have been related to the developed root structure and large proportion of roots in the deep soil (Table S1). Previous studies have indicated that competition between plants increases along with planting density. Likely to avoid competition between roots growing between rows [10,16], we here found that there was a large RWD at 11–22 cm from the center of narrow rows (Table S2). Especially under the higher planting density (D2), the increase in deep RWD of DH618 may allow the plants to seek additional soil resources. The increases in RLD in narrow rows may be due to the fact that the supply of water and nutrients were both in narrow rows. Under dense planting conditions, the effective response of the root structure in DH618 plants to the environment allowed the aboveground canopy to access more water and nutrients, delaying leaf senescence, prolonging the grain-filling time [23], and ultimately increasing grain yield.

Previous studies have shown that increasing maize density under limited light and assimilate conditions enables the root structure to develop a relatively large RL to cope with environmental changes [10]. In general, modern cultivars have higher RL than older cultivars [10,13]. Similarly, we here found that RL was significantly higher in the modern cultivar DH618 than in the older cultivar SC704, especially at the mature stage (Figure 4).



This may be due to the fact that DH618 had a compact plant type, which made more reasonable canopy structure and good green retention, allowing more nutrients to be allocated to the roots during the mature period [32,33]. Meanwhile, cultivar SC704 had a flat plant type, which reduced the light at lower canopy under high density, and then reduced the biomass accumulation and the allocation of nutrients to the roots during the mature period. Increased RL also provides a good source of nutrients for the growth and development of aerial plant tissue [13], especially during post-anthesis growth. Post-anthesis material accumulation has a higher contribution to yield than pre-anthesis accumulation [34,35]. Furthermore, increases in RL effectively increase the production of root exudates. Larger amounts of exudate can enable the root structure to penetrate more solid soil, allowing the roots to explore and manage more soil resources [11,36,37]. We here found that DH618 had significantly higher RL at a soil depth of 40–60 cm than SC704 did (Figure 4), indicating that DH618 could make use of more deep-soil resources; this is a very important way for plants to obtain more nutrients under limited resource input conditions, which plays a vital role in ensuring canopy growth and development [17].

Previous studies have shown no significant changes in RLD as planting density is increased [10,38]. However, our results showed significant increases in RLD at the silking and maturity stages as planting density increased (Table S4). This difference may be due to the lower planting densities used in previous studies [10]. When planting density is increased, maintenance of a high-quality larger canopy structure requires greater RLD; this ensures that maize plants can intercept water and nutrients more effectively during the growth period, which improves water and nutrient utilization efficiency and promotes high grain yield [16,39]. Here, DH618 had a larger RLD than SC704 (Table S4), which may be due to the fact that modern cultivars have more developed root structure under high density [24]. Furthermore, DH618 had a more compact root structure, which was conducive to the acquisition of soil resources in response to increased planting density [21,29]. A compact root structure effectively avoids resource competition between roots [10] and provides favorable conditions for exploring deep-soil resources [11]. This well-developed root structure may be one of the reasons underlying the good green retention and lack of premature senescence in the late stage observed in DH618; photosynthetic pigment synthesis requires N absorption and utilization, and photosynthesis requires water [40], and the soil structure of DH618 allowed abundant acquisition of N and water.

## 5. Conclusions

Cultivar DH618 had a higher grain yield than SC704 under high-density planting. RDW and RWD were 9.71% and 5.58% higher, respectively, in DH618 than in SC704. Furthermore, the RL distribution of DH618 was 6.82% higher at a soil depth of 20–60 cm compared to SC704, which promoted effective exploration of soil nutrients and provided sufficient nutrients for aerial tissues to function. Furthermore, RL and RLD were 20.92% longer and 16.97% larger, respectively, in DH618 than in SC704. These results indicated that DH618 had a more developed deep root system compared with SC704, which was beneficial for achieving high grain yield, especially under dense planting conditions.

**Supplementary Materials:** The following supporting information can be downloaded at: https://www.mdpi.com/article/10.3390/agriculture13040765/s1.

**Author Contributions:** Conceptualization, P.H. and S.L.; methodology, L.Z., G.L. and P.H.; software, L.Z.; validation, G.L., P.H. and S.L.; formal analysis, L.Z., G.L., Y.Y., J.X., B.M., R.X., K.W., S.L. and P.H.; investigation, L.Z., G.L., S.J., Y.Y. and X.G.; resources, P.H. and S.L.; data curation, L.Z.; writing-original draft preparation, L.Z. and G.L.; writing—review and editing, L.Z., G.L. and P.H.; visualization, L.Z., G.L. and P.H.; supervision, S.L. and P.H.; project administration, S.L.; funding acquisition, P.H. All authors have read and agreed to the published version of the manuscript.

**Funding:** This research was supported by the National Natural Science Foundation of China (32172118; 31871558), the National Key Research and Development Program of China (2016YFD0300110, 2016YFD0300101), the National Basic Research Program of China (973, Program 2015CB150401), the Basic Scientific Research Fund of Chinese Academy of Agricultural Sciences (S2021ZD05), the Agricultural Science and Technology Innovation Program (CAAS-ZDRW202004), the Central Public-interest Scientific Institution Basal Research Fund (No.S2022ZD05), and the Modern Agro-industry Technology Research System in China (CARS-02-25).

**Institutional Review Board Statement:** Not applicable.

**Informed Consent Statement:** Not applicable.

**Data Availability Statement:** The data are not publicly available due to privacy.

**Acknowledgments:** We are grateful to the reviewers and editors for helping us to improve the original manuscript.

**Conflicts of Interest:** The authors declare no conflict of interest.

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
