# Peer review of "Root Characteristics for Maize with the Highest Grain Yield Potential of 22.5 Mg ha−1 in China"

_agriculture, doi:10.3390/agriculture13040765_

Round 1

Reviewer 1 Report

Please do the following corrections:

-          In Abstract: Results did not contain narrow and/or wide rows cultivations. Please add these results to the abstract.

-          In Material and Methods: After the jointing stage, drip irrigation technology was used to conduct integrated quantitative water and fertilizer application every 10 d until plants were harvested. Fertilizer each 10 days is much more than needed. Please explain it. Irrigation each 10 days cannot be same intervals for during growing season because of different temperature and radiation during the season.

-          Please add the important dates of treatment, sampling and harvest in the form of calendar date, day after sowing and GDDs to the materials and methods.

-          In Results, Figures 3-4: Identify the significant symbols in the figures.

Reviewer 2 Report

The work is complex and interesting, but requires more explanations or clarifications. So:

-The title is not clearly formulated!!!! Repeat high-yield and yield potential !? It would not be recommended to write the value of the production in the title!!!!

The value of 22.5 Mg ha -1 is a sum of some hybrids, the average of the variants, a single hybrid?! Does it indicate biomass or grain production?

The production results were known in the working method!!! Then they are found in the results. When were they performed?

In conclusions:

- higher yield under high-density planting???

-higher yield of biomass or of grains?

Can the results of RDW and RWD be influenced by other factors? For example, the growing season? Are the two hybrids part of the same maturity group? But the genetic potential for growth and development of the hybrid?

See the manuscript!

Reviewer 3 Report

The manuscript “Root characteristics in high-yield maize with yield potential of 22.5 Mg ha-1” is a very interesting study. However, authors should improve some aspects :

1.       In the Results section, the Authors should review the subtitle 3.4 “As the planting density increased from D1 to D2, there was a corresponding significant increase in RWD (Table S3). In 2019, the average RWD of DH618 and SC704 increased from D1 to D2 by 36.15% and 10.51%, respectively; in 2021, the increases were 46.44% and 33.39%, respectively. Root length”. The subtitle is too long, possibly it is part of the results more than a title.

2.       In the Discussion section, the authors mention that the cultivar DH618 had the highest yield under the higher planting density; however, the results do not reflect this in the year 2021.

3.        In the Discussion section, the authors mention that cultivar DH618 is modern and that the genotype of the cultivar is very important for the characteristics that are observed in the roots. It would be important for the Authors to deepen both aspects in the discussion.

Round 2

Reviewer 2 Report

This paper ,,Root characteristics in high-yield maize with yield potential of 22.5 Mg ha-1", has many ambiguities regarding the links between production and the characteristics of the roots.

Why did DH618 not obtain a higher grain yield at a higher planting density in 2021?

Didn't DH618 achieve a higher grain yield at a higher planting density in 2021. In conclusion it is said that that DH618 had a more developed deep root system, which was beneficial for achieving high grain yield under dense planting conditions(It is contradictory in these explanations!)
